# The Serological Cross-Detection of Bat-Borne Hantaviruses: A Valid Strategy or Taking Chances?

**DOI:** 10.3390/v13071188

**Published:** 2021-06-22

**Authors:** Renata Carvalho de Oliveira, Jorlan Fernandes, Elba Regina de Sampaio Lemos, Fernando de Paiva Conte, Rodrigo Nunes Rodrigues-da-Silva

**Affiliations:** 1Laboratory of Hantaviroses and Rickettsioses, Oswaldo Cruz Institute, FIOCRUZ, Rio de Janeiro 21040-900, Brazil; reoliveira@ioc.fiocruz.br (R.C.d.O.); elemos@ioc.fiocruz.br (E.R.d.S.L.); 2Laboratory of Monoclonal Antibodies Technology, Immunobiological Technology Institute, FIOCRUZ, Rio de Janeiro 21040-360, Brazil; rodrigo.nunes@bio.fiocruz.br

**Keywords:** hantaviruses, bats, B-cell epitopes, bat-borne viruses

## Abstract

Bats are hosts of a range of viruses, and their great diversity and unique characteristics that distinguish them from all other mammals have been related to the maintenance, evolution, and dissemination of these pathogens. Recently, very divergent hantaviruses have been discovered in distinct species of bats worldwide, but their association with human disease remains unclear. Considering the low success rates of detecting hantavirus RNA in bat tissues and that to date no hantaviruses have been isolated from bat samples, immunodiagnostic tools could be very helpful to understand pathogenesis, epidemiology, and geographic range of bat-borne hantaviruses. In this sense, we aimed to identify in silico immunogenic B-cell epitopes present on bat-borne hantaviruses nucleoprotein (NP) and verify if they are conserved among them and other selected members of *Mammantavirinae*, using a combination of (the three most used) different prediction algorithms, ELLIPRO, Discotope 2.0, and PEPITO server. To support our data, we in silico modeled 3D structures of NPs from representative members of bat-borne hantaviruses, using comparative and ab initio methods due to the absence of crystallographic structures of studied proteins or similar models in the Protein Data Bank. Our analysis demonstrated the antigenic complexity of the bat-borne hantaviruses group, showing a low sequence conservation of epitopes among members of its own group and a minor conservation degree in comparison to *Orthohantavirus*, with a recognized importance to public health. Our data suggest that the use of recombinant rodent-borne hantavirus NPs to cross-detect antibodies against bat- or shrew-borne viruses could underestimate the real impact of this virus in nature.

## 1. Introduction

Bats (order Chiroptera) are taxonomically diverse, representing approximately 21% of all classified mammal species worldwide, with nearly 1422 species recognized [1]. The diversity of bats and their broad ecological features create many potential viral niches. In recent years, severe infectious diseases have been emerging, many of them caused by viruses originating from bats, such as Ebola virus, Marburg virus, Nipah virus, Hendra virus, and SARS coronaviruses [2]. Therefore, there is increasing evidence that bats are reservoirs for a high number of viruses, including hantaviruses [3].

Hantaviruses are important (re)emerging zoonotic pathogens previously classified in an unassigned order and in the genus *Hantavirus* of the family *Bunyaviridae*, recently reclassified and reassigned to a new family, designated *Hantaviridae*, in the order *Bunyavirales* [4,5]. The family *Hantaviridae* comprises four subfamilies: *Actantavirinae*, *Agantavirinae*, *Mammantavirinae*, and *Repantavirinae*. The subfamily *Mammantavirinae* has mammals, particularly rodents, as the most frequent natural hosts and is the largest and most complex *Hantaviridae* group, composed by four genera: *Loanvirus*, *Mobatvirus*, *Orthohantavirus*, and *Thottimvirus* [5]. Novel genotypes of hantaviruses have been identified in moles, shrews, and bats, expanding our understanding of the nature and origins of the hantaviruses [5,6,7,8]. Among the members of the subfamily *Mammantavirinae*, only *Orthohantavirus* is associated with human diseases: hemorrhagic fever with renal syndrome (HFRS) and hantavirus pulmonary syndrome, also called hantavirus cardiopulmonary syndrome (HPS or HCPS), respectively.

The presence of newly described hantaviruses in bats has challenged the conventional view that hantaviruses originated from rodents [8]. Several bat-borne hantaviruses are presently known, which show large genetic diversities from currently known rodent- and insectivore-borne (shrews and moles) hantaviruses [8,9,10]. Since the first report of hantavirus in *Nycteris hispida* and *Neoromicia nanus* bats in Africa in 2012, twelve bat-borne hantaviruses have been described in different regions of the world [3,11,12]. Of these, only *Laibin mobatvirus* (LAIV), *Longquan loanvirus* (LQUV), *Quezon mobatvirus* (QZNV), *Brno loanvirus* (BRNV), *Robina orthohantavirus* (ROBV), and *Xuan Son mobatvirus* (XSV) are currently recognized as species within genus *Mobatvirus* and *Loanvirus* by the International Committee on Taxonomy of Viruses (ICTV) released in 2020 [5]. Phylogenetic analysis of bat-borne hantavirus has indicated that bats might be the natural original hantavirus hosts. However, due to a lack of sufficient bat-borne hantavirus genomic sequences, their evolutionary phylogeny and genetic diversity as well as biological features are not fully understood [3,8,9,10,13,14].

Although bat-borne hantaviruses have been identified worldwide, the virus detection rate is low and restricted to limited locations, and it is still unclear if they cause infection or disease in humans. So far, none have been detected in the New World. Taking this into consideration, and the fact that none of these newfound bat-borne viruses have been isolated in cell culture in addition to the low success rates of detecting hantavirus RNA in bat tissues, improved diagnostic methods should be developed for virus detection [3,8]. In this context, serological tests could be important to uncover the real situation of bat-borne hantaviruses prevalence, help to understand their geographic distribution, and estimate the potential risk of these viruses to human health.

Nucleoprotein (NP) is the major antigen that elicits early serological responses in infected animals and has been used as a biomarker to develop antibodies for diagnostic and surveillance for several hantavirus species and genotypes [15,16,17]. To date, a few serological studies on bat-borne hantaviruses were conducted, using recombinant NP (rNP) of LAIV, XSV, and *Seoul orthohantavirus* (SEOV) as antigen for hantavirus detection in bat samples from China, and rNP of *Araraquara orthohantavirus* (ARAV rodent-borne hantaviruses) in bat samples from Brazil [18,19,20]. Therefore, this study aimed to verify if NPs B-cell epitopes of bat-borne hantaviruses and other selected members of *Mammantavirinae* are conserved among and between bat-, rodent-, and shrew-borne hantaviruses, using a combination of the three most used prediction algorithms, and to validate the epitopes predicted as specific to bat-borne viruses or as conserved among other hantaviruses.

## 2. Materials and Methods

### 2.1. Phylogenetic Analyses

The representative of complete NP sequences of hantavirus species as well as some unclassified hantaviruses [5] were downloaded from the GenBank^®^ database of NCBI (www.ncbi.nlm.nih.gov/nuccore; accessed on 28 October 2020). Multiple sequence alignment (MSA) of complete regions was performed using MAFFT version 7 (mafft.cbrc.jp/alignment/server; accessed on 28 October 2020). The best-fit evolutionary model was determined using MEGA version X, using the Bayesian Information Criterion [21]. We estimated phylogenetic relations of the NP sequences using maximum likelihood inference, as implemented in PhyML 3 [22] under the LG + G + I model of sequence evolution. Statistical support of the clades was measured by a heuristic search with 1000 bootstrap replicates in PhyML [23].

### 2.2. Amino Acid Composition Analysis

The full-length NP amino acid sequences from representative hantaviruses, such as, LAIV (YP_009507251), XSV (AHA83416), BRNV (APU53639), *Dakrong* virus (DKGV-AXQ03852), *Lena River* (LENV-QBH68029), LQUV (YP_009664869), *Nova mobatvirus* (NVAV-YP_009362031), and QZNV (YP_009361846) were obtained from the GenBank^®^ database of NCBI (www.ncbi.nlm.nih.gov/protein; accessed on 9 November 2020), as mentioned previously. Moreover, sequences of orthohantaviruses associated to rodent-borne diseases, including SEOV (ABM67564), *Hantaan orthohantavirus* (HTNV-ALA20831), *Andes orthohantavirus* (ANDV-AAK14323), ARAV (ABQ45402), and *Sin Nombre* (SNV-NP_941975) were involved in the study.

To analyze the amino acid composition of studied hantavirus NPs, we used the Composition Profiler Server (www.cprofiler.org/index.html; accessed on 9 November 2020), a web-based tool that automates detection of enrichment or depletion patterns of individual amino acids [24]. Hantavirus NPs amino acid compositions were compared using the SwissProt database [25]. Statistical significance associated with a specific enrichment or depletion is estimated using the two-sample t-test between two sequences of binary indicator variables, one sequence for each of the samples. A particular enrichment or depletion is statistically significant when the *p*-value (the lowest value at which the null hypothesis that the same underlying Gaussian distribution generated for both samples can be rejected) is lower than or equal to a user-specified statistical significance (alpha) value.

### 2.3. Hantavirus NP Amino Acid Variability Analysis

Sequences of studied hantavirus NPs were aligned using DNAStar Lasergene–Muscle Align (DNASTAR Inc., Madison, USA). The variability of amino acids in hantavirus NP sequences were evaluated using the Protein Variability Server (imed.med.ucm.es/PVS accessed on 17 November 2020). This server calculates the sequence variability within a multiple sequence alignment using Shannon entropy analysis [26]. Values of Shannon variability range from 0 (only one residue present at that position) to 4.322 (all 20 residues are equally represented in that position). Here, highly conserved positions are those with “Shannon variability” < 1.0. 

### 2.4. Analysis of Secondary Structure and Solvent Accessibility

The analysis of secondary structure and solvent accessibility of LAIV, XSV, BRNV, DKGV, LENV, LQUV, NVAV, and QZNV (representative hantaviruses from phylogroup II, see Results Section) and *orthohantavirus* associated to rodent-borne diseases (SEOV, HTNV, ANDV, ARAV, and SNV) were evaluated using the PredictProtein server (www.predictprotein.org; accessed on 9 December 2020) [27]. This server uses RePROFsec to predict secondary structure elements, such as helix, beta-strand, and loop. Furthermore, this server uses PROFacc to predict solvent accessibility of protein residues for 10 states of relative accessibility that are grouped into two main states: buried and exposed.

### 2.5. Three-Dimensional (3D) Structure Modeling

The 3D structures of selected NP from phylogroup II hantavirus were modeled using the Robetta server (new.robetta.org; accessed on 17 December 2021). This server is continually evaluated through CAMEO (Continuous Automated Model Evaluation) [28,29]. The quality of the protein structures generated was analyzed by MolProbity (molprobity.biochem.duke.edu; accessed on 6 January 2021) in order to choose the best predictive model generated for each NP. MolProbity is a widely used system of model validation for protein and nucleic acid structures [30]. Its unique feature of all-atom contact analysis (including hydrogens) was described in 1999 [31], followed by its complementary rotamer, Ramachandran, and Cβ deviation criteria [32,33], and the initial MolProbity web service [34].

### 2.6. Conformational B-Cell Epitope Prediction

After 3D protein structure modeling, structure-based discontinuous epitope prediction was initially performed using the ELLIPRO server (tools.iedb.org/ellipro; accessed on 12 January 2021), since it correlates epitope-antigenicity, solvent accessibility, and flexibility in antigen proteins for predicting conformational B-cell epitopes [4]. In addition to ELLIPRO analysis, we also ran the Discotope 2.0 server (tools.iedb.org/discotope; accessed on 15 January 2021), that uses a combination of amino acid statistics, spatial information, and surface exposure [5], and PEPITO server (pepito.proteomics.ics.uci.edu; accessed on 20 January 2021), that incorporates an amino acid propensity scale along with side-chain orientation and solvent accessibility information using half-sphere exposure values to predict conformational epitopes [6]. Potential conformational epitopes were considered only those that were more than 20% of Ellipro epitope predicted by all three algorithms, using their default thresholds.

### 2.7. Conservancy Analysis of the Selected Epitopes

To evaluate the conservancies of phylogroup II hantavirus epitopes among *Orthohantavirus* NPs used in serological tests, we used an approach based on the recent study of Ulah and collaborators [35]. Briefly, the IEDB server (http://tools.iedb.org/conservancy/; accessed on 22 January 2021) was used to determine the conservancy of the discontinuous epitopes setting the sequence identity threshold at ‘≥70’. Conservancy of epitopes can be defined as the fraction of protein sequences that contain the epitopes at/or above a certain level of identity [36,37]. Here, predicted epitopes were compared to *Orthohantavirus* commonly used to serologically cross-detect antibodies against *Mammantavirinae*: SEOV (ABM67564), HTNV (ALA20831), ANDVs (AAK14323), ARAV (ABQ45402), and SNV (NP_941975). 

## 3. Results

### 3.1. Phylogenetic Analyses

To construct phylogenetic relationships, 70 complete amino acid sequences of hantaviral NP were used. Currently, there are only seven complete NP sequences of bat-borne hantavirus species available in GenBank^®^. As shown in Appendix A, bat-, rodent-, and insectivore-borne hantaviruses showed a similar topology to the current taxonomic classification of *Hantavidae* family, here identified as clade I, II, III, and IV and V. Bat-borne hantaviruses were all clustered together within clade II, with two insectivore-borne hantaviruses, NVAV and LENV, respectively identified in *Talpa* moles and *Sorex* shrews, while other insectivore-borne hantaviruses were divided into clades I, II, III, IV, and VI. Only clade V contains solely rodent-borne hantaviruses, detected in the American continent.

### 3.2. NPs Vary in Length and Amino Acid Composition Profiles among Hantaviruses

In order to compare hantavirus NPs, Table 1 summarizes basic information about length and molecular weight of studied proteins in addition to ecological and taxonomic information about hantaviruses and their reservoirs. The length of the majority of hantavirus NPs ranges from 427 to 429 mers, however in phylogroup II, BRNV and LQUV were highlighted as shorter NPs, composed by 423 mers, while LENV presented the longer protein sequence, composed by 448 mers. Besides the length variation, hantavirus NPs presented similar molecular weight, ranging from ARAV 47,878.46 Da to LENV 50,532.14 Da.

Regarding the amino acid composition of hantavirus NPs, we observed a diverse composition profile among studied proteins (Figure 1). Using the Composition Profiler server, we compared the amino acids of hantavirus NP with the SwissProt database and identified enriched and depleted amino acids in each studied protein. In our study, only ANDV, ARAV, HTNV, and LAIV did not present any enriched or depleted amino acids. On the other hand, methionine residues (M) were enriched in SEOV (*p* = 0.01), NVAV (*p* = 0.01), QZNV, and DKGV (*p* = 0.01). DKGV also presented enriched leucine residues (*p* = 0.02), such as XSV (*p* = 0.03), ROBV (*p* = 0.03), and QZNV (*p* = 0.01), that additionally presented enriched aspartate residues (*p* = 0.01). Moreover, BRNV presented enriched glutamate (*p* = 0.01) and arginine (*p* = 0.05) residues and depleted alanine residues (*p* = 0.02), while LQUV presented enriched glutamine residues (*p* = 0.02) and depleted asparagine residues (*p* = 0.04). This asparagine residue was also depleted in LENV NP (*p* = 0.02), which yet presented enriched aspartate (*p* = 0.04) and glutamine (*p* = 0.04) residues. Besides, glycine residues were depleted in SNV (*p* = 0.02) and ROBV (*p* = 0.02) NPs.

### 3.3. C-Terminal Region of NPs Were More Conserved among the Studied Hantavirus

As illustrated in Figure 2, hantavirus NPs presented about 44% of amino acid sequence conservancy with the consensus. Moreover, the C-terminal region of NPs, presenting 65% of highly conserved amino acids, was more conserved among the studied hantavirus than the N-terminal region. C-terminal and N-terminal regions presented 65% and 48% of amino acids highly conserved, respectively.

### 3.4. Hantavirus NPs Share Similar Secondary Structures and Solvent Accessibility

Based on the amino acid sequences of studied proteins, we analyzed the secondary structure and solvent accessibility of hantavirus NPs. Despite the low conservation degree, we observed a similar profile of secondary structures and proportion of exposed amino acids in solvent. In all studied hantavirus NPs, we identified about 50% of amino acids within helix, 5% of amino acids as β-strand, and 45% as loop secondary structure (Appendix A). Regarding the accessibility of amino acids in solvent, in all studied proteins, 53% to 56% of amino acids were predicted as exposed in solvent, while the proportion of buried amino acids in solvent ranged from 39% to 41% (Appendix A).

### 3.5. Validation of Phylogroup II Hantavirus NPs Models

The quality of the modeled proteins’ structures was evaluated and validated using the MolProbity server, based on the Ramachandran plot, that described the amino acid positions in the plot as well as the overall [38], and in the complementary rotamer, Ramachandran, and Cβ deviation criteria [32,33]. Here, in all selected models, the plot showed that more than 97.2% of amino acids were arranged in Ramachandran favored zones and presented more than 98.7% favored rotamers. Moreover, predicted models for LAIV, NVAV, QXNV, XSV, and DKGV NPs did not present outliers in the Ramachandran plot, only one was presented for BRNV and LQUV and two for ROBV and LENV. In the same way, among the used hantavirus NPs models, the LAIV model presented one poor rotamer, while all other models did not present poor rotamers (Table 2). These data showed that the generated structures of phylogroup II hantavirus NPs were good considering the overall geometry and can be used for further study.

### 3.6. Predicted Conformational B-Cell Epitopes in Hantavirus NPs Models

After validating and evaluating the quality of phylogroup II hantavirus NPs models and aiming to predict its main B-cell epitopes, we initially used Ellipro to predict discontinuous epitopes in combination with Discotope 2.0 and Pepito to refine the analysis. In this way, sixty epitopes were predicted by Ellipro, however, only 55% (33 epitopes) of these epitopes were also predicted by Discotope 2.0 and Pepito (Appendix A). Conformational epitopes lengths ranged from 4 to 118 mers. As showed in Figure 3, LQUV, LAI, QZNV, and ROBV NPs presented three predicted epitopes, while four epitopes were predicted in BRNV, NVAV, XSV, and DKGV models, and five were predicted in the LENV virus NP model. Moreover, all predicted epitopes are located in exposed regions of hantavirus NPs and thus can be recognized by specific antibodies (Figure 3b,d,f,h,j,l,n,p,r).

### 3.7. Most of Conformational B-Cell Epitopes Are Not Conserved among Hantavirus

Aiming to verify the conservancy of predicted B-cell epitopes among hantaviruses, they were compared to phylogroup II hantavirus NPs (BRNV, LQUV, LAIV, NVAV, QZNV, XSV, LENV, DKGV, and ROBV). Besides, considering the use of *Orthohantavirus* NPs for serological detection of bat-borne hantavirus, these epitopes were also compared to ANDV, ARAV, SEOV, SNV, and HTNV NPs. Despite the high conservancy observed among the C-terminal region of hantavirus NPs, most of the predicted NPs B-cell epitopes were not conserved among phylogroup II hantavirus and *Orthohantavirus* enrolled in this study (Table 3). Focusing on the epitopes’ conservancy among phylogroup II hantavirus, only XSV and DKGV viruses presented 50% (2/4) of their epitopes (XS2, XS3, D1, and D4), showing a high average degree of conservation (≥70%) among the majority of representative phylogroup II hantaviruses, while BRNV, LAIV, QZNV, LENV, and ROBV NPs presented only one epitope highly conserved among this phylogroup. Moreover, BRNV, LAIV, XSV, and LENV viruses NPs presented one epitope (B4, LB3, XS3, and LS4, respectively) highly conserved (identity ≥ 70%) among 93% (13/15) of studied hantaviruses (Table 4). On the other hand, LQUV and NVAV did not present any highly conserved epitope among studied viruses.

## 4. Discussion

Several bat-borne hantaviruses have recently been identified, but to date, it is not clear whether they are pathogenic for humans. Thus far, a total of 12 hantaviruses were identified in 16 different species of Old-World bats in Africa, Asia, Europe, and Australia [3,39]. The discovery of highly divergent lineages of hantaviruses in bats of different species and their vast geographic distribution provides unlimited opportunities to search for other bat-associated hantaviruses, including in the New World. However, the low success rates of detecting hantavirus RNA in bat tissues observed in previous studies and the difficulty to isolate hantavirus from bat samples [3] reinforce the need to develop alternative and more sensitive diagnostic tools to help in understanding the role of hantaviruses harbored by bats on disease, epidemiology, and their geographic range.

In this sense, immunodiagnostic tools could be very helpful to uncover the real scenario of bat-borne hantaviruses, that is still poorly understood. Remarkably, NP is the major hantavirus antigen that elicits early humoral immune responses and has been used as a serological target and for antigen detection in animals and patients [40,41,42]. This protein contains both serotype-specific and common group epitopes [43,44,45] and has been used to investigate the presence of hantaviruses in bats [18] and shrews [46] and to screen bats that could harbor hantaviruses in the New World [19,20]. However, there are no studies comparing the similarity of B-cell epitopes among bat- and rodent-borne hantaviruses.

Here, our data suggest that the general tertiary structures of NP seem to be similar among members of the subfamily *Mammantavirinae*, since all studied proteins presented similar frequencies of exposed regions and secondary structures (Figure 2). Nevertheless, NPs of bat- and shrew-borne hantaviruses from phylogroup II were more variable in length and molecular weight than the ones belonging to *Orthohantavirus*, with lengths ranging from 423 to 448 mers and about 48,630 Da of mean molecular weight, while *Orthohantavirus* NP presented about 428 amino acids of length and 48,035 Da of mean molecular weight (Table 1). Moreover, we observed non-conserved profiles of amino acid composition among studied hantaviruses NPs, in which only ANDV, ARAV, HNTV, and LAIV did not present any enriched or depleted amino acid residue (Figure 1). These data are in agreement with the analysis of conservation degree among studied proteins that showed about 51% of amino acids’ sequence conservation. Epitopes can be distributed over the whole NP, but it is thought that the C-terminal half of NP contains rather conformation- and multimerization-dependent epitopes, which should be more serotype-specific [47,48]. In a recently published work from our group, we demonstrated that NP amino acid sequence conservation among *Orthohantavirus* associated to HFRS (SEOV, HNTV, *Amur virus*, *Dobrava-Belgrade orthohantavirus*, and *Puumala orthohantavirus*) and associated to HCPS (ANDV, SNV, and *Laguna Negra orthohantavirus*) ranged from 62% to 95% [45]. The N-terminal quarter of hantaviral NP bears linear and immunodominant epitopes, but as seen here, a possible and interesting antigenic site was found in the C-terminal half [47]. Thus, we believe that the observed low conservation among NPs from phylogroup II hantavirus in comparison with NPs from representative *Orthohantavirus* associated to rodent-borne disease corroborates the view that identification of common immunodominant epitopes in these proteins is critical for a better selection of targets to serological detection of antibodies.

Recently, Xu and collaborators [18], using recombinant NP from phylogroup II hantaviruses (LAIV and XSV) and SEOV, observed the cross-reaction of XSV-infected bat serum samples with SEOV rNP, while samples of bats infected by LAIV were not able to cross-react to SEOV rNP [18]. Indirectly, this finding makes us question whether the use of *Orthohantavirus* nucleoproteins cannot underestimate the presence of antibodies against bat-borne hantavirus, especially in the New World, where the knowledge about bat-borne hantavirus species and their reservoirs remains scarce. Considering this, our aim in this study was to predict immunodominant B-cell epitopes in phylogroup II hantavirus NPs and to compare them among phylogroup II and also among representative *Orthohantavirus* associated to rodent-borne diseases classically used in serological antibody detection.

B-cell epitope prediction aims to facilitate the identification of antigenic regions of proteins, that could be used to replace antigens for antibody production and to accelerate the development of novel vaccines and diagnostic/serologic tools [49]. In this context, conformational B-cell epitopes correspond to about 90% of protein surface-accessible clusters of amino acids able to be recognized by secreted antibodies or B-cell receptors eliciting humoral immune response [50]. Unfortunately, its prediction is limited by the knowledge of protein 3D structure, information only available for a few proteins [51], and absolutely scarce to hantavirus proteins. Despite the absence of crystallographic structures of studied proteins or similar models in the Protein Data Bank [52], we in silico modeled 3D structures of NPs from representative members of phylogroup II hantaviruses (BRNV, LQUV, LAIV, NVAV, QZNV, XSV, LENV, DKGV, and ROBV), using comparative and ab initio methods. All generated structures were well-evaluated (Table 2) and considered good models for conformational B-cell epitopes’ predictions [30,53,54,55].

Thus, in each studied NP, between three to five clusters of amino acids were predicted as exposed discontinuous B-cell epitopes, composed by 4 to 118 mers in each cluster (Figure 3). From our point of view, the similar number of predicted epitopes in studied proteins corroborates the similar profiles of exposed amino acids among NP hantavirus (data not shown). Moreover, the number of amino acids in predicted epitopes ranged from 30% to 44% of the protein length, and we believe that this variation was acceptable due to differences in protein length (Table 1) and 3D structure models. Besides, our data corroborates the study of Kalaiselvan and collaborators, who reported a similar number of B-cell linear epitopes among *Orthohantavirus* associated to HFRS [43]; however, this was the first study aiming to identify and to compare B-cell epitopes among phylogroup II hantaviruses.

Regarding the conservation of hantavirus NP-exposed regions, our observations showed that the majority of predicted epitopes are not conserved among hantavirus phylogroup II, since only 9 (27%) of 33 predicted epitopes share more than 70% of their amino acids with at least half of the selected representative members of this group. Interestingly, among studied representative members of phylogroup II hantavirus, LQUV and NVAV did not present conserved epitopes with other members of this group, while XSV and DKGV viruses presented two out of four highly conserved predicted epitopes. Remarkably, our data suggested that XSV NP is antigenically closer to other phylogroup II hantaviruses, since it shares three highly conserved epitopes with LQUV and LAIV, two epitopes with QZNV, DKGV, and ROBV viruses, and one epitope with BRNV and with hantaviruses harbored by shrews (NVAV and LENV). Considering the geospatial context, the geographic distribution of their reservoir hosts and their overlapping areas with other bat species (where hantavirus host-switching likely occurs), could also be taken into account in the serological surveillance of hantavirus regionally. XSV and DKGV were reported in bat species that belong to the family Hipposideridae, one of the most speciose of insectivorous bats. The reservoir hosts of XSV, the Pomona roundleaf bats (*Hipposideros pomona*) and Ashy roundleaf bats *(Hipposideros cineraceus*), have a higher geographic distribution area in comparison to the Stoliczka’s Asian trident bats (*Aselliscus stoliczkanus*), the reservoir host of DKGV. The vast geographic distribution of both of these Hipposideros species in Asia, that are sympatric and often roost in the same caves, and the distribution of 70 other species members of this large genus, provide opportunities to detect XSV-related hantaviruses or other genetically divergent mobatviruses throughout Africa, Europe, Asia, and Australia [3]. To date, only one study used two different recombinant proteins of phylogroup II hantavirus for serological detection of hantaviruses in bats, demonstrating a high reactivity among studied samples against XSV and LAIV recombinant NPs [18].

On the other side, recombinant *Orthohantavirus* NPs have been used to screen antibodies against bat-borne hantaviruses. Recently, recombinant SEOV NP was used to cross-detect antibodies against LAIV and XSV [18] and recombinant ARAV NP was used to investigate whether phyllostomid bats could harbor hantaviruses in Brazil [20]. Our data demonstrated that phylogroup II hantavirus NPs are antigenically low-conserved among *Orthohantavirus* (Table 4), presenting no more than one highly conserved B-cell epitope shared with the majority of compared *Orthohantavirus* (ANDV, ARAV, SNV, SEOV, and HTNV). Based on our findings, the use of recombinant *Orthohantavirus* NPs to cross-detect antibodies against bat-borne hantaviruses could lead to an underestimation of the real reactivity, resulting in low-sensibility strategies, since only seven (21%) of phylogroup II hantavirus NP predicted epitopes were highly conserved among *Orthohantavirus* NP. Corroborating with the findings of Xu and collaborators [18], who reported cross-reactivity among XSV and SEOV recombinant NP but observed no cross-reactivity against LAIV, another bat-borne hantavirus, and SEOV NPs. Here, we demonstrated that XSV NP shares two epitopes with SEOV, while LAIV shares only one short epitope, explaining the absence of cross-reactivity. Moreover, similar results were found by Wei at al. in their studies with shrew-borne hantavirus, which suggested low or no cross-reactivity among *Seewis orthohantavirus*, *Altai orthohantavirus*, *Thottapalayam thottimvirus*, *Asama orthohantavirus*, and rodent-borne hantaviruses [56]. Thereby, we believe that the knowledge about the conservation of natural epitopes is critical for serological diagnosis based on cross-reactions, especially considering the limited information about the spread of bat-borne hantaviruses around the world [3,39,57]. This suggestion is supported by Tischler’s study, that showed that ANDV NP presented different humoral key targets to humans and to rodents [58], reinforcing the critical role of conserved epitopes to higher chances of cross-reaction.

Efforts to develop novel and accurate diagnostic tools able to detect bat-borne hantaviruses are urgent to allow a better comprehension of its spread and its real impact on animal and human health. Here, we demonstrated the antigenic complexity of bat-borne hantaviruses, showing a low sequence conservation of epitopes among members of its own group and a minor conservation degree in comparison with rodent- and shrew-borne hantaviruses. Our data suggest that the use of recombinant rodent-borne hantavirus NPs to cross-detect antibodies against bat- or shrew-borne viruses could underestimate the real impact of this virus in nature. Diagnostic use of homologous virus antigen in the assay seems to be preferred to search for antibodies after infection by a particular hantavirus. For a broader detection, among studied proteins, XSV NP presented the higher sequence conservation among phylogroup II hantavirus members, and we believe that should be considered the better choice for serological cross-detection of bat-borne hantavirus. Besides, immunodominant linear B-cell epitopes of bat-borne hantavirus NPs can allow the development of novel and specific diagnostic approaches, but this strategy still needs to be explored by accurately identifying antibody targets.

## Figures and Tables

**Figure 1 viruses-13-01188-f001:**
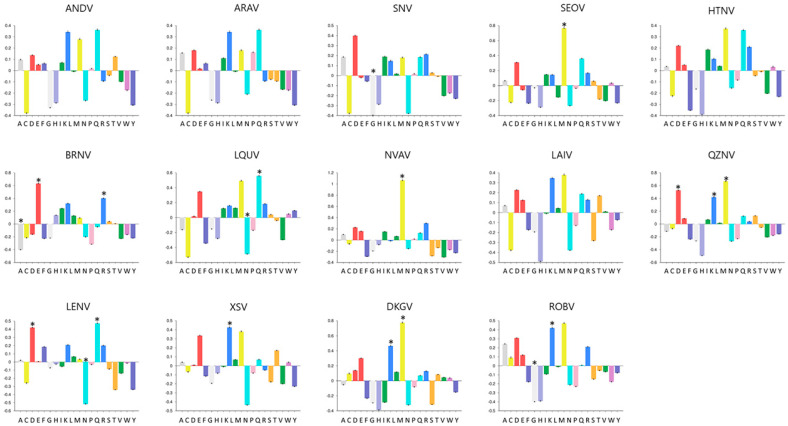
Composition profiles of hantavirus NPs. Amino acids are represented on the *x*-axis: alanine (A; dark grey bar), cysteine (C; yellow bar), aspartate (D; bright red bar), glutamate (E; bright red bar), phenylalanine (F; mid blue bar), glycine (G; light grey bar), histidine (H; pale blue bar), isoleucine (I; green bar), lysine (K; blue bar), leucine (L; green bar), methionine (M; yellow bar), asparagine (N; cyan bar), proline (P; flesh bar), glutamine (Q; cyan bar), arginine (R; blue bar), serine (S; orange bar), threonine (T; orange bar), valine (V; green bar), tryptophan (W; pink bar), tyrosine (Y; mid blue bar). Depleted and/or enriched amino acids were indicated by *. Statistical significance associated with a specific enrichment or depletion is estimated using the two-sample t-test between two sequences of binary indicator variables, one sequence for each of the samples. A particular enrichment or depletion is statistically significant when the *p*-value (the lowest value at which the null hypothesis that the same underlying Gaussian distribution generated for both samples can be rejected) is lower than or equal to a defined statistical significance value (*p* = 0.05).

**Figure 2 viruses-13-01188-f002:**
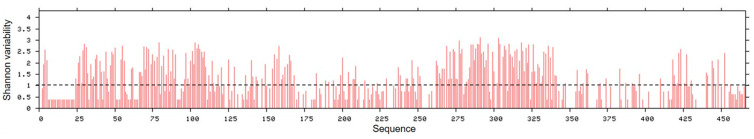
Analysis of hantavirus NPs variability. The variability analyses were evaluated comparing *Laibin mobatvirus* (YP_009507251), *Xuan Son virus* (AHA83416), *Brno loanvirus* (APU53639), *Dakrong virus* (AXQ03852), *Lena River virus* (QBH68029), *Longquan loanvirus* (YP_009664869), *Nova mobatvirus* (YP_009362031), *Quezon mobatvirus* (YP_009361846), *Seoul orthohantavirus* (ABM67564), *Hantaan orthohantavirus* (ALA20831), *Andes orthohantavirus* (AAK14323), *Araraquara virus* (ABQ45402), and *Sin Nombre orthohantavirus* (NP_941975) NPs with consensus sequence. The graph *y*-axis represents the Shannon entropy. The traced line indicates the threshold value of conservation, and values ≤ 1.0 indicate highly conserved positions.

**Figure 3 viruses-13-01188-f003:**
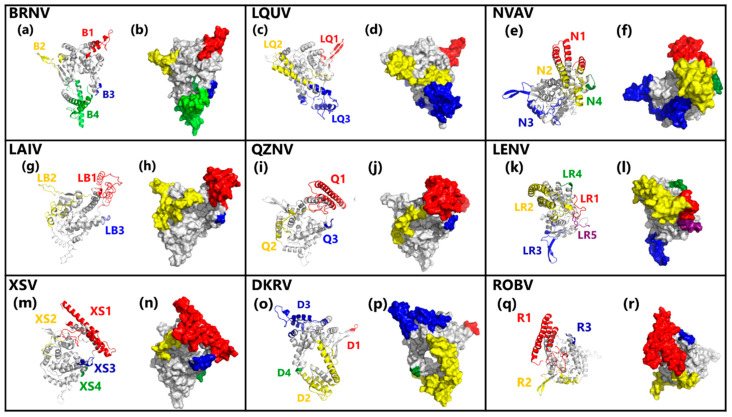
Mapping of predicted B-cell conformational epitopes is highlighted on the hantavirus NPs models. Mapping of: (**a**,**c**,**e**,**g**,**i**,**k**,**m**,**o**,**q**) secondary structural elements and (**b**,**d**,**f**,**h**,**j**,**l**,**n**,**p**,**r**) surface accessibility.

**Table 1 viruses-13-01188-t001:** General information of studied hantaviruses and their NPs.

Taxonomic Data			Ecological Data				Nucleocapsid Protein Data	
Hantavirus	Phylogroup	Genus	Reservoir Order	Reservoir Families	Reservoir Species	Virus Distribution	Length	Molecular Weight (Da)
Andes virus (ANDV)	V	*Orthohantavirus*	Rodentia	Cricetidae	*Oligoryzomys longicaudatus*	Argentina, Chile	428	48,039.69
Araraquara virus (ARAV)	V	*Orthohantavirus*	Rodentia	Cricetidae	*Necromys lasiurus*	Brazil	428	47,878.46
Sin Nombre virus (SNV)	V	*Orthohantavirus*	Rodentia	Cricetidae	*Peromyscus maniculatus*	USA, Canada, and Mexico	428	48,174.57
Seoul virus (SEOV)	IV	*Orthohantavirus*	Rodentia	Muridae	*Rattus rattus* and *R. norvegicus*	Worldwide	429	47,930.31
Hantaan virus (HTNV)	IV	*Orthohantavirus*	Rodentia	Muridae	*Apodemus agrarius*	China, Korea, Japan, and Russia	429	48,152.61
Brno virus (BRNV)	II	*Loanvirus*	Chiroptera	Vespertilionidae	*Nyctalus noctula*	Czech Republic	423	48,699.39
Longquan virus (LQUV)	II	*Loanvirus*	Chiroptera	Rhinolophidae	*Rhinolophus sinicus; R. affinis* and *R. monoceros*	China	423	48,193.69
Nova virus (NVAV)	II	*Mobatvirus*	Soricomorpha	Talpidae	*Talpa europaea*	Belgium, France, Hungary, and Poland	428	48,418.26
Laibin virus (LAIV)	II	*Mobatvirus*	Chiroptera	Emballonuridae	*Taphozous melanopogon*	China, Myanmar	427	48,108.92
Quezon virus (QZNV)	II	*Mobatvirus*	Chiroptera	Pteropodidae	*Rousettus amplexicaudatus*	Philippines	429	48,441.9
Lena River virus (LENV)	II	*Mobatvirus*	Soricomorpha	Soricidae	*Sorex caecutiens*	Russia	448	50,532.14
Xuan Son virus (XSV)	II	*Mobatvirus*	Chiroptera	Hipposideridae	*Hipposideros pomon;* *H. cineraceus*	Vietnam	427	48,136.93
Dakrong virus (DKGV)	II	*Unclassified*	Chiroptera	Hipposideridae	*Aselliscus stoliczkanus*	Vietnam	427	48,612.78
Robina virus (ROBV)	II	*Orthohantavirus*	Chiroptera	Pteropodidae	*Pteropus alecto*	Australia	429	48,525.26

**Table 2 viruses-13-01188-t002:** Geometry analysis of hantavirus NPs models.

Hantavirus	Poor Rotamers	Favored Rotamers	Ramachandran Outliers	Ramachandran Favored	Z-Score
**Brno**	0	377	1	412	1.15 ± 0.39
	0.00%	99.21%	0.24%	97.86%	
**Longquan**	0	369	1	414	0.66 ± 0.40
	0.00%	99.73%	0.24%	98.34%	
**Nova**	0	365	0	418	0.88 ± 0.38
	0.00%	99.46%	0.00%	98.12%	
**Laibin**	1	364	0	413	0.59 ± 0.38
	0.27%	99.18%	0.00%	97.18%	
**Quezon**	0	375	0	420	1.04 ± 0.38
	0.00%	99.47%	0.00%	98.36%	
**Lena River**	0	378	2	438	−0.08 ± 0.37
	0.00%	98.69%	0.45%	98.21%	
**Xuan Son**	0	367	0	418	0.20 ± 0.37
	0.00%	99.73%	0.00%	98.35%	
**Dakrong**	0	370	0	418	0.43 ± 0.38
	0.00%	98.93%	0.00%	98.35%	
**Robina**	0	369	2	420	0.79 ± 0.39
	0.00%	100%	0.47%	98.36%	
**Goals**	<0.3%	>98%	<0.05%	>98%	<2

**Table 3 viruses-13-01188-t003:** Analysis of epitopes’ conservation of hantavirus phylogroup II NPs.

Virus	Epitope	Length	Conservation among *Orthohantavirus*		Conservation among Phylogroup II	
			High Similar NPs (≥70%) (%—N/5)	Mean Conservation % (Min–Max)	High Similar NPs (≥70%) (%—N/9)	Mean Conservation % (Min–Max)
**BRNV**	B1	35	0.00 (0/5)	21.1 (20–22.9)	11.11 (1/9)	35.9 (20–100)
	B2	28	0.00 (0/5)	40 (35.7–42.9)	11.11 (1/9)	50.4 (21.4–100)
	B3	103	0.00 (0/5)	22.1 (20.4–24.3)	11.11 (1/9)	35.5 (21.4–100)
	B4	7	100.00 (5/5)	80 (71.4–85.7)	88.89 (8/9)	92.1 (57.1–100)
**LQUV**	LQ1	25	0.00 (0/5)	54.4 (48–60)	11.11 (1/9)	52.4 (44–100)
	LQ2	37	0.00 (0/5)	22.7 (18.9−29.7)	11.11 (1/9)	36.6 (18.9–100)
	LQ3	88	0.00 (0/5)	44.5 (42–46.6)	11.11 (1/9)	55.6 (40.9–100)
**NVAV**	N1	44	0.00 (0/5)	35.5 (22.7–43.2)	11.11 (1/9)	31.8 (18.2–100)
	N2	53	0.00 (0/5)	38.9 (32.1–45.3)	11.11 (1/9)	42.6 (7.6–100)
	N3	57	0.00 (0/5)	47 (43.9–49.1)	11.11 (1/9)	57.3 (31.6–100)
	N4	9	0.00 (0/5)	53.3 (44.4–55.6)	11.11 (1/9)	66.7 (55.6–100)
**LAIV**	LB1	72	0.00 (0/5)	27.5 (26.4–29.2)	22.22 (2/9)	53.1 (33.3–100)
	LB2	65	0.00 (0/5)	39.7 (35.4–44.6)	33.33 (3/9)	60.9 (40–100)
	LB3	8	100.00 (5/5)	80 (75–87.5)	88.89 (8/9)	93.1 (62.5–100)
**QZNV**	Q1	95	0.00 (0/5)	14.1 (10.5–18.9)	22.22 (2/9)	29.5 (8.4–100)
	Q2	32	0.00 (0/5)	55 (50–56.3)	22.22 (2/9)	62.5 (37.5–100)
	Q3	6	40.00 (2/5)	73.3 (66.7–83.3)	88.89 (8/9)	90.7 (50–100)
**LENV**	LR1	10	0.00 (0/5)	50 (50–50)	11.11 (1/9)	61.9 (57.1–100)
	LR2	85	0.00 (0/5)	10.4 (9.4–11.8)	11.11 (1/9)	20.7 (7.1–100)
	LR3	36	0.00 (0/5)	25.6 (22.2–30.6)	11.11 (1/9)	30.6 (19.4–100)
	LR4	4	80.00 (4/5)	80 (50–100)	100.00 (9/9)	77.8 (75–100)
	LR5	12	0.00 (0/5)	41.7 (41.7–41.7)	11.11 (1/9)	42.6 (33.3–100)
**XSV**	XS1	99	0.00 (0/5)	20.8 (18.9–24.2)	22.22 (2/9)	48.4 (19.2–100)
	XS2	15	40.00 (2/5)	65.3 (60–73.3)	77.78 (7/9)	75.6 (46.7–100)
	XS3	11	100.00 (5/5)	80 (72.7–81.82)	88.89 (8/9)	79.8 (54.6–100)
	XS4	4	0.00 (0/5)	50 (50–50)	22.22 (2/9)	58.3 (50–100)
**DKGV**	D1	5	0.00 (0/5)	52 (40–60)	55.56 (5/9)	77.8 (60–100)
	D2	118	0.00 (0/5)	28.6 (26.3–30.5)	33.33 (3/9)	49.4 (17.8–100)
	D3	59	0.00 (0/5)	16.3 (15.3–18.6)	11.11 (1/9)	40.1 (15.3–100)
	D4	6	40.00 (2/5)	73.3 (66.7–83.3)	88.89 (8/9)	90.7 (50–100)
**ROBV**	R1	102	0.00 (0/5)	15.4 (11.1–20.2)	22.22 (2/9)	31.6 (13.1–100)
	R2	63	0.00 (0/5)	45.1 (42.9–47.6)	22.22 (2/9)	59.4 (31.8–100)
	R3	6	40.00 (2/5)	73.4 (66.7–83.3)	88.89 (8/9)	90.7 (50–100)

**Table 4 viruses-13-01188-t004:** Conservation degree of hantavirus phylogroup II NPs conformational epitopes.

Virus	Epitope	Length	Conservation Degree (%)													
			Rodent-Borne					Bat-Borne							Shrew-Borne	
			ANDV	ARAV	SNV	SEOV	HTNV	BRNV	LQUV	LAIV	QZNV	XSV	DKGV	ROBV	LENV	NVAV
**Brno**	B1	35	20	20	23	20	23	100	54	20	26	23	29	26	26	20
	B2	28	43	43	43	36	36	100	57	43	43	39	54	57	21	39
	B3	103	20	21	20	24	24	100	43	22	25	21	27	24	30	26
	B4	7	86	86	71	86	71	100	100	100	100	86	100	100	86	57
**Longquan**	LQ1	25	48	48	56	60	60	56	100	44	52	44	48	60	24	44
	LQ2	37	19	19	19	27	30	22	100	41	24	43	35	24	19	22
	LQ3	88	47	47	42	43	44	69	100	47	50	41	48	51	52	42
**Nova**	N1	44	43	43	43	25	23	32	27	18	23	18	20	27	20	100
	N2	53	45	43	42	32	32	8	13	49	51	47	42	45	28	100
	N3	57	49	49	44	47	46	44	47	56	60	60	61	56	32	100
	N4	9	56	56	56	44	56	56	67	67	67	56	67	67	56	100
**Laibin**	LB1	72	28	26	26	28	29	21	28	100	57	78	69	60	32	33
	LB2	65	37	38	35	43	45	40	45	100	57	82	82	62	29	52
	LB3	8	88	88	75	75	75	100	100	100	100	88	100	100	88	63
**Quezon**	Q1	95	11	11	12	19	19	9	8	15	100	14	14	71	22	13
	Q2	32	50	50	56	56	63	53	56	53	100	66	59	81	38	56
	Q3	6	83	83	67	67	67	100	100	100	100	83	100	100	83	50
**Lena River**	LR1	10	50	50	50	50	50	57	57	57	57	57	57	57	100	57
	LR2	85	9	9	11	12	11	7	11	11	13	9	12	13	100	11
	LR3	36	22	22	25	28	31	25	22	19	22	19	22	22	100	22
	LR4	4	75	75	100	50	100	75	75	75	75	75	75	75	100	75
	LR5	12	42	42	42	42	42	33	42	33	33	33	33	42	100	33
**Xuan Son**	XS1	99	20	19	18	22	24	19	29	78	40	100	70	43	22	33
	XS2	15	60	60	60	73	73	67	73	80	80	100	87	73	47	73
	XS	11	82	82	73	82	82	73	82	91	82	100	82	82	73	55
	XS4	4	50	50	50	50	50	50	75	50	50	100	50	50	50	50
**Dakrong**	D1	5	60	60	60	40	40	60	60	100	80	100	100	60	60	80
	D2	118	29	28	26	30	31	18	20	75	51	70	100	48	25	37
	D3	59	19	17	15	15	15	24	25	46	34	47	100	47	15	22
	D4	6	83	83	67	67	67	100	100	100	100	83	100	100	83	50
**Robina**	R1	102	11	12	13	20	20	13	13	17	73	17	16	100	22	13
	R2	63	43	44	48	44	46	51	48	60	73	59	63	100	32	49
	R3	6	83	83	67	67	67	100	100	100	100	83	100	100	83	50

The epitopes’ degree of conservation among studied hantaviruses were indicated in the table (%) and classified as low conservancy (light-blue cells; conservancy < 40%), intermediate conservancy (yellow cells, ≥40% conservancy < 70%), and high conservancy (red cells, conservancy ≥ 70%). The columns represent hantavirus species: *Andes orthohantvirus* (ANDV); *Araraquara virus* (ARAV); *Sin Nombre orthohantavirus* (SNV); *Seoul orthohantavirus* (SEOV); *Hantaan orthohantavirus* (HTNV); *Brno loanvirus* (BRNV); *Longquan loanvirus* (LQUV); *Nova mobatvirus* (NVAV); *Laibin mobatvirus* (LAIV); *Quezon mobatvirus* (QZNV); *Lena River mobatvirus* (LENV); *Xuan Son virus* (XSV); *Dakrong virus* (DKGV); *Robina orthohantavirus* (ROBV).

## Data Availability

Data is contained within article and Appendix A.

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
