# Peer review of "The Serological Cross-Detection of Bat-Borne Hantaviruses: A Valid Strategy or Taking Chances?"

_viruses, 2021, doi:10.3390/v13071188_

Round 1
Reviewer 1 Report
In their ms, Oliveira et al., reported the possibility of the nucleocapsid proteins for the serodiagnosis of bat-borne hantaviruses. I am convinced that the study should be of interest for the field of hantavirology but I have a few comments in the research. Please see below the points.
Authors should choose more reasonable title as the research article.
Authors provided figures 1 and 3 for analysis. Are there any significance information from those figures? Authors should remove those figures.
Authors should provide conclusion for this research.
Author Response
Reviewer 1
In their ms, Oliveira et al., reported the possibility of the nucleocapsid proteins for the serodiagnosis of bat-borne hantaviruses. I am convinced that the study should be of interest for the field of hantavirology but I have a few comments in the research. Please see below the points.
We appreciate the valuable suggestions of the reviewers and present below the appropriateness and/or justification point by point below.
Authors should choose more reasonable title as the research article.
Answer: The authors are grateful for your comments on our manuscript, we have chosen the current title, based on previous studies that have used rodent-borne hantavirus antigens to investigate hantavirus infections in bats, with no cross-reaction evaluation, what could be a valid strategy or tanking chances. That’s why we would like to keep the current title.
Authors provided figures 1 and 3 for analysis. Are there any significance information from those figures? Authors should remove those figures.
Answer: Thank you for the suggestion, we agree that figure 3 could be moved to supplementary material, but the authors consider that Figure 1, comparison of amino acid profile composition between different hantavirus nucleoproteins, is important to discuss the role of conformational B-cell epitope differences among them. In this sense, Figure 1 can help readers to better understand why there are few conformational epitopes shared between these viruses. Taking into consideration that, by a biochemical point of view, significant structural and conformational changes could be associated to amino acid composition of nucleoproteins and it could be the reason for not using Orthohantavirus NP as antigen for phylogroup II serologic detection.
Authors should provide conclusion for this research.
Answer: The following sentence (L459 – L473) is what the authors consider to be their conclusion: “Efforts to develop novel and accurate diagnostic tools able to detect bat-borne hanta-viruses are urgent to allow a better comprehension of its spread and its real impact in animal and human health. Here we demonstrated the antigenic complexity of bat-borne hantaviruses, showing a low sequence conservation of epitopes among members of its own group and a minor conservation degree in comparison with rodent- and shrew-borne hantaviruses. Our data suggest that the use of recombinant rodent-borne hantavirus NPs to cross detect antibodies against bats- or shrew–borne viruses could underestimate the real impact of this virus in nature. Diagnostic using of homologous virus antigen in the assay seems to be preferred to search for antibodies after infection by a particular hanta-virus. For a broader detection, among studied proteins, XSV NP presented the higher sequence conservation among phylogroup II hantavirus members and we believe that should be considered the better choice to serological cross detection of bat-borne hanta-virus. Besides, immunodominant linear B-cell epitopes of bat-borne hantavirus NPs can allow the development of novel and specific diagnostic approaches, but this strategy still needs to be explored by accurate identifying antibody targets.”
Reviewer 2 Report
The authors use a series of in silico analyses to model structural and biochemical scenarios of Mammntavirinae (chiropteran-borne) B-cell epitope nucleoproteins (NP), to better understand sequence conservation. They provide valuable insight into a largely understudied group of hantavirus reservoirs and discuss the importance of underestimated occurrence of hantaviruses in the environment largely attributed to common surveillance methods.
Major concerns:
Tables: I could find no overall rationale in the organization of data presented in the tables and figures (e.g., alphabetical, taxon-level, distribution, etc.). For example, Figure 1 could be arranged with viruses listed alphabetically or in descending order of phylogenetic similarity. Another example would be to organize the table according to taxonomy or molecular wt., etc.
More specifically: to me, Table 1 is very important to your overall analyses in terms of host evolution framed in a geospatial context. In actuality, the distributions presented in the table are grossly underestimated. Perhaps the authors may have listed where the samples were collected? I would highly recommend updating the distribution columns for each reservoir species. A strong source can be used from (https://www.iucnredlist.org/) or using Wilson and Reeder (2005) Mammal Species of the World (available online: https://www.departments.bucknell.edu/biology/resources/msw3/) in conjunction with the asm database the authors provide. This is important because it supports my argument against the choice of XSV as a focus for cross-detections. The reason for disagreement is related to the overlapping distributions of reservoirs sharing, or potentially sharing an area with multiple competent host taxa. The generality of a detection method could be implied regionally or provide focused detection methods where hantavirus spillover likely occurs.
For example, the authors show the distribution of XSV to Vietnam and carried by Hipposideros spp. The actual distribution of these species as far west as India. This is important because of the regional overlap with reservoirs known to carry highly cross-reactive hantaviruses. One could argue that DKGV would be more appropriate due to the similarity of epitope structural qualities with XSV, yet the distribution of its reservoir host has a higher distribution. Another example could be NVAV, where host-switching is well known. Perhaps the authors could strengthen the argument for XSV as a choice for surveillance-detection where very little of the statistical variation is explained.
Fig 2: This is a very nice figure however I found my version very difficult to read. Again, the rationale for the arrangement is confusing. Also, it would be helpful to see a subtle horizontal line at the Shannon threshold (~1.0) across all sequence values.
Minor revisions:
L36: “originating”
L39-(throughout): taxonomic families are generally not italicized (unless under journal guidelines).
L44: ”most complex”(?)
L51: “. . . HCPS, respectively.”
L57: “hantaviruses”
L63: “hantavirus”
L66-75: Paragraph is confusing; suggest re-organize.
L76: Suggest changing the first word of the sentence.
L91: “hantaviruses”
L97: “. . . evolution. Statistical support . . .”
L105: “. . . orthohantaviruses associated with rodent-borne . . .”
L109-114: Sentences are a little confusing; suggest reorganize.
L139: “Robetta’s”(?)
L157-159: Why 20%?; maybe provide a description of the threshold choice.
L178: “Talpa” & “Sorex” genus italics
L184: “. . . and their reservoirs.”
L190: “Length” in column of Table 1
L194-throughout: p-values could be reduced to 2 decimal places (0.05)
L203: “Amino acids are represented . . “
L214: what was your alpha value?
L239: “struc3ture”? typo? or program?; not sure.
L248: “. . . (%) of amino . . .”
L261-264: a little confusing; suggest reorganization.
L280: “. . . (Figure 4b,d,f,etc. . . .)”
L286: “low”?
L344: “. . . are in agreement. . . “
L354-358: supports distribution as a predictor.
L364: agreed; the authors’ contribution could be applied to New World bats.
L381: suggest using different word to open first full sentence.
L392: “hantaviruses”
L394: “. . . the majority of . . “
L395: “once”?
L400-406: supports the importance of reservoir distribution sharing as a predictor.
415-416: this reviewer completely agrees with the authors
L422: “found”
Author Response
Reviewer 2
The authors use a series of in silico analyses to model structural and biochemical scenarios of Mammantavirinae (chiropteran-borne) B-cell epitope nucleoproteins (NP), to better understand sequence conservation. They provide valuable insight into a largely understudied group of hantavirus reservoirs and discuss the importance of underestimated occurrence of hantaviruses in the environment largely attributed to common surveillance methods.
We appreciate the valuable suggestions of the reviewers and present below the appropriateness and/or justification point by point below.
Major concerns:
Tables: I could find no overall rationale in the organization of data presented in the tables and figures (e.g., alphabetical, taxon-level, distribution, etc.). For example, Figure 1 could be arranged with viruses listed alphabetically or in descending order of phylogenetic similarity. Another example would be to organize the table according to taxonomy or molecular wt., etc.
Answer: Virus presentation and organization in all tables and figures were reconsidered and are now presented by phylogroup and follow the order presented in Table 1.
More specifically: to me, Table 1 is very important to your overall analyses in terms of host evolution framed in a geospatial context. In actuality, the distributions presented in the table are grossly underestimated. Perhaps the authors may have listed where the samples were collected? I would highly recommend updating the distribution columns for each reservoir species. A strong source can be used from (https://www.iucnredlist.org/) or using Wilson and Reeder (2005) Mammal Species of the World (available online: https://www.departments.bucknell.edu/biology/resources/msw3/) in conjunction with the asm database the authors provide. This is important because it supports my argument against the choice of XSV as a focus for cross-detections. The reason for disagreement is related to the overlapping distributions of reservoirs sharing, or potentially sharing an area with multiple competent host taxa. The generality of a detection method could be implied regionally or provide focused detection methods where hantavirus spillover likely occurs.
Answer: We agree, that host distribution is very important, but this has already be done by Arai S and Yanagihara R. (Genetic Diversity and Geographic Distribution of Bat-borne Hantaviruses. Curr Issues Mol Biol. 2020; 39:1-28), so in table 1 we opted to inform only the location (country) where the hantaviruses have been found in their reservoir host, excepted by SEOV, since it has a wide geographic distribution. To clarify this information we change the headline in Table 1 to “Virus distribution” instead of “World distribution”.
For example, the authors show the distribution of XSV to Vietnam and carried by Hipposideros spp. The actual distribution of these species as far west as India. This is important because of the regional overlap with reservoirs known to carry highly cross-reactive hantaviruses. One could argue that DKGV would be more appropriate due to the similarity of epitope structural qualities with XSV, yet the distribution of its reservoir host has a higher distribution. Another example could be NVAV, where host-switching is well known. Perhaps the authors could strengthen the argument for XSV as a choice for surveillance-detection where very little of the statistical variation is explained.
Answer: Thank you for the suggestion, we include the following information about their geographic distribution in the text: “XSV and DKGV were reported in bat’s species that belongs to the family Hipposideridae, one of the most speciose of insectivorous bats. The reservoir hosts of XSV, the Pomona roundleaf bats (Hipposideros pomona) and Ashy roundleaf bats (Hipposideros cineraceus) have a higher geographic distribution area in comparison to the Stoliczka’s asian trident bats (Aselliscus stoliczkanus), the reservoir host of DKGV. The vast geographic distribution of these both Hipposideros species in Asia, that are sympatric and often roost in the same caves, and the distribution of other 70 species members of this large genus provide opportunities to detect XSV-related hantaviruses or other genetically divergent mobatviruses throughout Africa, Europe, Asia and Australia. Arai S and Yanagihara R. 2020”
Fig 2: This is a very nice figure however I found my version very difficult to read. Again, the rationale for the arrangement is confusing. Also, it would be helpful to see a subtle horizontal line at the Shannon threshold (~1.0) across all sequence values.
Answer: We agree with your suggestion, and we added a line at the Shannon threshold. We also change the original figure and kept only Figure 2a.
Minor revisions:
L36: “originating”
Answer: The term was rewritten as suggested.
L39-(throughout): taxonomic families are generally not italicized (unless under journal guidelines).
Answer: Accordioning to ICTV code like a species name, a higher taxon name is written in italics and begins with a capital letter, that’s why the authors decided to keep viral family names in italic. https://talk.ictvonline.org/information/w/faq/386/how-to-write-virus-species-and-other-taxa-names
L44: ”most complex”(?)
Answer: The term was replaced by “most frequent”.
L51: “. . . HCPS, respectively.”
Answer: The term was added to the text as suggested.
L57: “hantaviruses”
Answer: The term was rewritten as suggested.
L63: “hantavirus”
Answer: The term was rewritten as suggested.
L66-75: Paragraph is confusing; suggest re-organize.
Answer: The paragraph was re-organized as suggested: “Although bat-borne hantaviruses have been identified worldwide, the virus detection rate is low and restricted to limited locations and it’s still unclear if they cause infection or disease in humans. So far, none have been detected in the New World. Taking it into consideration, and the fact that none of these newfound bat-borne viruses have been isolated in cell culture in addition to the low success rates of detecting hantavirus RNA in bat tissues, improved diagnostic methods should be developed for virus detection [3,8,].”
L76: Suggest changing the first word of the sentence.
Answer: The word was replaced as suggested.
L91: “hantaviruses”
Answer: The term was rewritten as suggested.
L97: “. . . evolution. Statistical support . . .”
Answer: The phrase was rewritten as suggested.
L105: “. . . orthohantaviruses associated with rodent-borne . . .”
Answer: The term was rewritten as suggested.
L109-114: Sentences are a little confusing; suggest reorganize.
Answer: The paragraph was re-organized as suggested: “Hantavirus NPs aminoacid compositions were compared using SwissProt database”
L139: “Robetta’s”(?)
Answer: The term was rewritten as suggested.
L157-159: Why 20%?; maybe provide a description of the threshold choice.
Answer: The use of twenty percent is based on our previous unpublished data. Prediction of conformational epitopes using Ellipro generates larges epitopes when compared to other algorithms. In order to increase our success rate on B cell epitope prediction, we arbitrarily choose to consider a “good” epitope, sequences that shares more than 20% of Ellipro epitope predicted by all three algorithms.
L178: “Talpa” & “Sorex” genus italics
Answer: The term was rewritten as suggested.
L184: “. . . and their reservoirs.”
Answer: The term was rewritten as suggested.
L190: “Length” in column of Table 1
Answer: The term was rewritten as suggested.
L194-throughout: p-values could be reduced to 2 decimal places (0.05)
Answer: All p-values were reduced to two decimal places.
L203: “Amino acids are represented... “
Answer: The term was rewritten as suggested.
L214: what was your alpha value?
Answer: Alpha value is the statistical significance value used in this study (p=0.05). Considering the comment, we modify the text “(…) is lower than or equal to a defined statistical significance value (p=0.05).”
L239: “struc3ture”? typo? or program?; not sure.
Answer: It was typo error the word was corrected in the text.
L248: “. . . (%) of amino . . .”
Answer: The word added to the text as suggested.
L261-264: a little confusing; suggest reorganization.
Answer: The paragraph was re-organized as suggested: “Moreover, predicted models for LAIV, NVAV, QXNV, XSV, and DKGV NPs did not presented outliers in Ramachandran plot, only one was presented for BRNV and LQUV and two for ROBV and LENV”
L280: “. . . (Figure 4b,d,f,etc. . . .)”
Answer: The term was corrected as suggested.
L286: “low”?
Answer: The term was replaced by: “not conserved among hantavirus”.
L344: “. . . are in agreement. . . “
Answer: The term was rewritten as suggested.
L354-358: supports distribution as a predictor.
Answer: Thank you for the comment.
L364: agreed; the authors’ contribution could be applied to New World bats.
Answer: Thank you for the comment.
L381: suggest using different word to open first full sentence.
Answer: The term was rewritten as suggested.
L392: “hantaviruses”
Answer: The term was rewritten as suggested.
L394: “. . . the majority of . . “
Answer: The term was rewritten as suggested.
L395: “once”?
Answer: The term was replace by since.
L400-406: supports the importance of reservoir distribution sharing as a predictor.
Answer: Thank you for the comment.
415-416: this reviewer completely agrees with the authors
Answer: Thank you for the comment.
L422: “found”
Answer: The term was rewritten as suggested.